# Proposal of a New Definition of “Very Early” Intrahepatic Cholangiocarcinoma—A Retrospective Single-Center Analysis

**DOI:** 10.3390/jcm10184073

**Published:** 2021-09-09

**Authors:** Oliver Beetz, Angelica Timrott, Clara A. Weigle, Andreas Schroeter, Sebastian Cammann, Juergen Klempnauer, Florian W. R. Vondran, Felix Oldhafer

**Affiliations:** Department of General, Visceral and Transplant Surgery, Hannover Medical School, Carl-Neuberg-Str. 1, 30625 Hannover, Germany; Beetz.Oliver@mh-hannover.de (O.B.); Timrott.Angelica@mh-hannover.de (A.T.); Clara.A.Weigle@stud.mh-hanover.de (C.A.W.); Andreas.Schroeter@stud.mh-hannover.de (A.S.); Cammann.Sebastian@mh-hannover.de (S.C.); Klempnauer.Juergen@mh-hannover.de (J.K.); Vondran.Florian@mh-hannover.de (F.W.R.V.)

**Keywords:** intrahepatic cholangiocarcinoma, early disease, prognostic factors, postoperative morbidity, postoperative survival

## Abstract

Intrahepatic cholangiocarcinoma (ICC) is a rare disease with poor outcome, despite advances in surgical and non-surgical treatment. Recently, studies have reported a favorable long-term outcome of “very early” ICC (based on tumor size and absence of extrahepatic disease) after hepatic resection and liver transplantation, respectively. However, the prognostic value of tumor size and a reliable definition of early disease remain a matter of debate. Patients undergoing resection of histologically confirmed ICC between February 1996 and January 2021 at our institution were reviewed for postoperative morbidity, mortality, and long-term outcome after being retrospectively assigned to two groups: “very early” (single tumor ≤ 3 cm) and “advanced” ICC (size > 3 cm, multifocality or extrahepatic disease). A total of 297 patients were included, with a median follow-up of 22.8 (0.1–301.7) months. Twenty-one (7.1%) patients underwent resection of “very early” ICC. Despite the small tumor size, major hepatectomies (defined as resection of ≥3 segments) were performed in 14 (66.7%) cases. Histopathological analyses revealed lymph node metastases in 5 (23.8%) patients. Patients displayed excellent postoperative outcome compared to patients with “advanced” disease: intrahospital mortality was not observed, and patients displayed superior long-term survival, with a 5-year survival rate of 58.2% (versus 24.3%) and a median postoperative survival of 62.1 months (versus 25.3 months; *p* = 0.013). In conclusion, although the concept of a “very early” ICC based solely on tumor size is vague as it does not necessarily reflect an aggressive tumor biology, our proposed definition could serve as a basis for further studies evaluating the efficiency of either surgical resection or liver transplantation for this malignant disease.

## 1. Introduction

Intrahepatic cholangiocarcinoma (ICC) is a highly malignant disease with poor prognosis and the second most common primary hepatic malignancy, responsible for 10–20% of all primary liver tumors [1,2].

Recent epidemiological data revealed a rising incidence; however, ICC remains a rarity, especially compared to hepatocellular carcinoma [2]. Although there is a lack of large clinical studies to identify patients, which benefit most from surgical resections, several prognostic factors for postoperative survival have been identified in the past, including multifocality, vascular invasion, lymph node involvement, and type of liver resection [3,4,5,6,7]. The tumor diameter itself as a prognostic factor has been a matter of debate: while the first staging system of the American Joint Committee on Cancer (AJCC) did not include tumor size, the 8th edition of the AJCC(/UICC) defines stage IA and IB depending on a maximum tumor size of 5 cm [8,9]. Furthermore, the Liver Cancer Study Group of Japan demonstrated that a tumor cut-off diameter of 2 cm had a significant prognostic value in 419 patients undergoing resection of ICC [10]. Recently, several reports were published illustrating the excellent long-term outcome of patients with a so-called “very early” ICC (single tumor ≤ 2 cm, absence of extrahepatic disease) undergoing liver transplantation for advanced cirrhosis and hepatic failure [11,12]. However, the majority of affected patients are not eligible for transplant. Data of patients with “very early” ICC undergoing surgical resection are scarce, especially as patients often present themselves with more advanced disease. Molina et al. identified seven patients meeting the above-mentioned criteria for “very early” ICC in preoperative imaging in their study of 89 patients undergoing liver resection with an improved 5-year survival rate of 68.6% but failed to show statistically significant superior outcome [13].

As the incidence of patients with single tumors ≤2 cm is extremely low and data presented by Spolverato et al. demonstrated the prognostic value of a 3 cm tumor size cut-off as a predictor for vascular invasion and higher tumor grading in patients undergoing resection [14], while Sapisochin et al. concluded in their work that 3 cm could prove as a more appropriate cut-off regarding transplant eligibility [11], we aimed to evaluate the outcome of patients undergoing resection of ICC depending on a tumor size of ≤3 cm and to thereby further define the concept of a “very early” disease.

## 2. Materials and Methods

### 2.1. Study Design

We retrospectively reviewed 297 patients undergoing resection of ICC in curative intent between February 1996 and January 2021 at the Department of General, Visceral and Transplant Surgery, Hannover Medical School, in Hannover, Germany, for postoperative morbidity, mortality, and long-term survival. Patients were included only if conclusive postoperative histology was available. Resections of combined hepatocellular–cholangiocarcinomas, perihilar cholangiocarcinomas, and gallbladder carcinomas were not included within the analyses. Patients undergoing surgery for tumor recurrence and patients under the age of 18 years were also excluded from this study. 

### 2.2. Definition of “Very Early” and “Advanced” Intrahepatic Cholangiocarcinoma

Patients were assigned to two groups on the basis of tumor size, multifocality, and extrahepatic disease: “very early” ICC was defined by the presence of a single tumor with a size ≤3 cm and the absence of extrahepatic disease. All other patients were assigned to the group of “advanced” ICC.

### 2.3. Definition of Variables

Major hepatectomies were defined as resections of three or more liver segments, whereas extended hepatectomies were defined as resection of five or more segments. Vascular resections were defined as simultaneous resection of large vessels, such as the vena cava inferior or the portal vein. Multivisceral resections were defined as additional resections of extrahepatic tissue (excluding perihilar bile ducts and large vessels). Postoperative complications were graded according to the Clavien-Dindo classification system as follows: Grade 0 (no complications), grade I (minor deviations), grade II (requiring pharmacologic treatment), grade III (requiring interventions), grade IV (life-threatening), and grade V (death). Complications ≥grade III were defined as severe complications. In addition, the classification system of the International Study Group of Liver Surgery for common surgical complications post hepatic resection (biliary leakage, hemorrhage, and liver failure) was applied. The AJCC/UICC staging system (8th edition) was applied for tumor classification. The follow-up time was defined as time between hepatic resection and last contact or death. The last follow-up date was 17 May 2021. Survival times are reported as Kaplan–Meier median estimates throughout the text.

### 2.4. Statistical Analysis

The distribution of categorical variables between the two groups was compared by the chi-squared and Fisher’s exact test. Differences in continuous data were analyzed with the Student’s t test in case of normal distribution and the Mann-Whitney U test. Kaplan–Meier analyses and log rank tests were applied to compare postoperative survival between the two groups. Statistical significance was set at a *p*-value < 0.050 and is shown in bold (Tables) or marked with an asterisk (Figure). The collected data were implemented and analyzed using SPSS statistical software (version 26; SPSS Inc.; IBM Corporation, Armonk, NY, USA). The figure was created with GraphPad Prism (version 9.1.0 for Windows, GraphPad Software, La Jolla, CA, USA).

## 3. Results

### 3.1. Patient Cohort

A total of 297 patients was included in the analyses, with a median follow-up of 22.8 (0.1–301.7) months. Twenty-one (7.1%) patients underwent surgical resection for “very early” ICC, whereas 276 (92.9%) patients were resected for “advanced” ICC. Table 1 summarizes selected variables and a statistical comparison of the two groups. Except for significantly lower preoperative leukocyte concentrations and a higher rate of male patients, biometrical data and preoperative laboratory values did not differ significantly in patients undergoing resection of “very early” ICC when compared to all other patients. Of note, except for a lack of “very early” ICC resected between 1996 and 2000, we did not observe a trend towards an increased detection and resection of these “very early” tumors in recent years, despite advances in diagnostic imaging (Table 2).

### 3.2. Surgical Details

Despite the small tumor size in patients with “very early” ICC, 14 (66.7%) patients underwent major hepatectomies, including 5 (23.8%) patients receiving simultaneous perihilar bile duct resection and 1 (4.8%) patient undergoing extended hepatectomy. Major resections despite a “very early” disease were performed due to a tumor location adjacent to either central vascular structures or bile ducts. One patient underwent left hemihepatectomy as a radical resection due to intraoperatively suspected nodal involvement. Although pedicle clamping during resection was performed in 16 (76.2%) patients, intraoperative transfusions were necessary in 6 (28.6%) cases (versus *n* = 135 (48.9%); *p* = 0.071). However, the extent of (hepatic) surgery, vascular and extrahepatic resections, and consequently operating times were inferior by trend when compared to that in patients undergoing resection of “advanced” ICC (see Table 1).

### 3.3. Histopathological Results

Postoperative histopathological analyses revealed lymph node metastases in five (23.8%) cases, comparable to the rate of patients suffering from “advanced” ICC (*n* = 80, 29.0%; *p* = 0.777). Of note, the rate of simultaneous lymphadenectomy was also similar in the two groups (61.9% versus 65.2%; *p* = 0.814). 

Vascular invasion was identified less frequently for “very early” disease (*n* = 1 (4.8%) versus *n* = 66 (23.9%); *p* = 0.020), and none of the patients revealed a poorly differentiated carcinoma (versus *n* = 83 (30.1%); *p* = 0.005).

Furthermore, positive resection margins were observed in 1 patient (4.8%) only, as opposed to 45 (16.3%) patients undergoing resection for “advanced” ICC (*p* = 0.218). Table 1 summarizes the histopathological results for both groups, including a statistical comparison.

### 3.4. Postoperative Course and Survival

Patients undergoing resection for “very early” ICC showed superior postoperative outcome, defined by morbidity and mortality. Severe complications were observed in four (19.0%) patients only (versus *n* = 97 (35.1%); *p* = 0.156). None of the patients suffered from severe post-hepatectomy liver failure. Moreover, intrahospital, 30-day, or 90-day mortality was not observed in the case of “very early” disease, whereas 21 (7.6%; *p* = 0.379) patients died within the primary hospital stay after undergoing resection of “advanced” disease. The main cause for intrahospital mortality was a postoperative liver failure with subsequent multiple organ failure, observed in nine (42.9%) cases. Six (28.6%) patients died due to septic complications (of pulmonary, abdominal, or unknown origin). Three (14.3%) patients died after massive postoperative hemorrhage. Lethal myocardial infarction was observed in two (9.5%) patients, whereas mesenteric ischemia was determined as the cause of death in one (0.5%) patient. Table 3 displays selected postoperative outcome variables of both groups, including a statistical comparison. At the time of the last follow-up, 15 (71.4%) patients with “very early” disease were deceased. Survival analyses revealed 1-, 3-, and 5-year survival rates of 95.2%, 79.4%, and 58.2% and a significantly superior median survival of 62.1 months, when compared to all other patients (25.3 months; *p* = 0.013, Figure 1).

## 4. Discussion

Despite efforts to improve prognostic stratification for patients undergoing resection of ICC, preoperatively available und reliable tumor-related factors are still lacking. The prognostic value of the tumor size, which—in times of modern medical imaging—is easily determinable prior resection, is surprisingly vague. Recently, the concept of a “very early” ICC (single tumor ≤ 2 cm) has been introduced, displaying excellent outcome after liver transplantation and resection [11,12,13]. The definition of “very early” ICC is based on a survey of the Liver Cancer Study Group of Japan including 27 patients, who enjoyed a remarkable 5-year survival of 82.4% after liver resection, and has received positive appraisal in a multicenter study evaluating liver transplantation in the context of ICC by Sapisochin et al., showing a favorable 5-year survival of 65.0% [10,11]. However, the latter study was not able to demonstrate an independent prognostic value of the tumor size on disease recurrence and the authors expressed doubts about the proposed cut-off of 2 cm, recommending a prospective trial with tumors of up to 3 cm [11]. Molina et al. retrospectively identified seven patients meeting the criteria for “very early” ICC in preoperative imaging (although only five patients fulfilled the criteria in the final histopathological examination) and reported an excellent 5-year survival rate of 68.6% after resection but failed to show a statistically significant superior outcome [13]. Although the small sample sizes as a result of the low incidence of patients with tumors ≤2 cm, as presented by Molina and Sapisochin et al., make it difficult to demonstrate statistical significance, the available data suggesting discrimination of a “very early” disease for a tumor size of 2 cm are not convincing. Spolverato et al. were further able to demonstrate that patients with a tumor smaller than 3 cm had a lower incidence of microscopic vascular invasion and high-grade tumors, even when compared to patients with tumors between 3 and 5 cm in a study including 443 patients [14]. For these reasons, we aimed to characterize patients with single tumors ≤3 cm undergoing resection in this retrospective analysis and to thereby provide a clinically more relevant definition of a “very early” ICC.

In our cohort of 297 patients who underwent surgical resection at a tertiary referral center for hepatobiliary surgery over a course of 25 years, we identified 21 patients suffering from “very early” ICC. These patients displayed excellent postoperative outcome and long-term survival, with a 5-year survival rate 58.2%, which was significantly superior to that of patients with more advanced disease. Furthermore, we were able to confirm the aforementioned data from Spolverato et al. with significantly lower rates of vascular invasion and high-grade tumors. In the light of our own results and other work on the matter, we conclude that a 3 cm tumor size cut-off seems to be more appropriate for the definition of a “very early” ICC and could serve as a basis for further studies evaluating the efficiency of either surgical resection or liver transplantation and also for interventional approaches such as thermal ablation, as was recently reported by Kim et al., for this malignant disease [15].

Nevertheless, the value of tumor size in general for long-term prognosis remains uncertain. Studies of the past failed to show a direct correlation between tumor size and survival. Consequently, the first ICC-specific classification according to the AJCC/UICC staging system (7th edition) did not include tumor size [16]. 

More recently, Hyder et al. demonstrated an independent prognostic value of tumor size for survival after resection of ICC in a patient cohort of 514 patients [4]. Interestingly, a tumor diameter of more than 7 cm did not further increase the hazard ratio of mortality. 

In response to the aforementioned reports, also including studies by Ali et al. and Doussot et al., the current (8th) edition of the AJCC/UICC staging system subdivides the T1 stage with regard to a maximum tumor size of more than 5 cm (a/b) [8,17,18].

As has been concluded by others, a larger tumor size is often associated with further pathologic features impairing the long-term outcome of patients, such as perforation of the visceral peritoneum and, thereby, even involvement of extrahepatic structures, which are also factors implemented within the 8th edition of the AJCC/UICC classification system, but also more oblique factors not part of the current staging systems, such as an obstruction of the biliary tree.

Recently, inflammation-based biomarkers have been shown to have prognostic value in patients with ICC [19]. Accordingly, we previously identified preoperative leukocytosis as an independent risk factor for survival after resection of ICC and proposed malnutrition and dehydration, cholangitis due to cholestasis, as well as a systemic reaction to a more aggressive tumor biology as explanations for our findings [5]. In line with these results, we observed significantly lower levels of preoperative leukocyte concentrations in patients undergoing resection of “very early” ICC. On the contrary, 23.8% of these patients were found to have nodal involvement, clearly reflecting a more advanced disease and most probably an aggressive tumor biology. Aside from these deliberations, the comparatively high rate of lymph node metastases in patients with “very early” disease underlines the importance of regional lymphadenectomy not only for a correct staging according to the current AJCC/UICC staging system but also for further therapeutic decision making [20].

In the past years, the understanding of cancer biology has made significant progress, and mutation analyses of tumor tissue as well as modern biomarkers could prove more appropriate to predict prognosis and response to potential therapies in patients with ICC, although the available data are still scarce and partially conflicting. In two large studies, mutations in KRAS and TP53 were associated with inferior recurrence-free and overall survival [21,22]. Furthermore, Ruys et al. published a large meta-analysis with 4126 patients demonstrating that the biomarkers fascin, EGFR, MUC1, and MUC4 are associated with reduced survival in patients with resected ICC [23].

In summary, we believe that, despite our encouraging results, the concept of a “very early” ICC merely defined by tumor size is vague, as it does not necessarily exclude an aggressive tumor biology and an advanced disease. Therefore, further studies evaluating the efficiency of either surgical resection or liver transplantation and also interventional approaches should be based on modern tumor biomarkers or mutation analyses in combination with inflammation-based biomarkers and traditional surgical risk factors, such as tumor size, to optimize the outcome of this lethal disease.

Limitations of our study are the retrospective design as well as the long time period of more than two decades in which the included patients underwent surgical resection. Furthermore, the low incidence and therefore the small number of cases with “very early” ICC requires multicenter studies to address pending issues in the future.

## Figures and Tables

**Figure 1 jcm-10-04073-f001:**
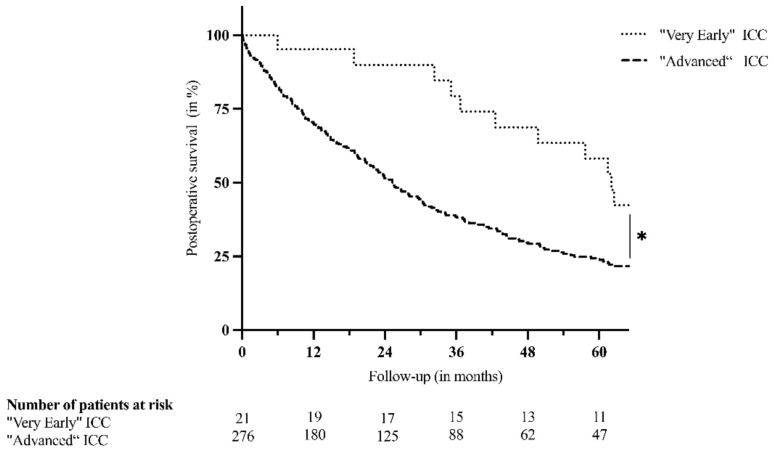
Postoperative survival after resection of “very early” (single tumor ≤ 3 cm) and “advanced” (>3 cm, multifocality or extrahepatic disease) intrahepatic cholangiocarcinoma (ICC). * *p*-value: 0.013.

**Table 1 jcm-10-04073-t001:** Perioperative variables of patients with “very early” intrahepatic cholangiocarcinoma (ICC) including a comparison to a control group undergoing resection with curative intent.

Variables	“Very Early” ICC (Single Tumor ≤ 3 cm; *n* = 21)			“Advanced” ICC (*n* = 276)			*p*-Value
	*n* (%)	Mean, Median (Range)	M.v. (*n* (%))	*n* (%)	Mean, Median (Range)	M.v. (*n* (%))	
**Biometrics**							
Age (in years)		62.8, 62 (45–78)	0 (0)		61.6, 62 (24–84)	0 (0)	0.728
Male	17 (81.0)		0 (0)	138 (50.0)		0 (0)	**0.006**
BMI (in kg/m^2^)		25.9, 25.2 (19.2–38.2)	0 (0)		25.8, 25.2 (16.4–55.4)	5 (1.8)	0.962
**Preoperative data**							
Hemoglobin (in g/dL)		13.6, 13.5 (10.7–16.4)	0 (0)		13.2, 13.4 (8.2–17.2)	1 (0.4)	0.359
Leukocytes (×10^3^/µL)		6.8, 6.5 (3.6–14.4)	0 (0)		8.4, 7.9 (1.7–24.1)	1 (0.4)	**0.003**
Platelets (×10^3^/µL)		257.5, 247 (134–495)	0 (0)		280.5, 255 (69–902)	2 (0.7)	0.322
Quick (in %)		98.9, 101 (67–120)	0 (0)		99.1, 100 (46–147)	3 (1.1)	0.924
ASAT (in U/L)		36.3, 30.5 (10–140)	1 (4.8)		39.5, 31 (4–304)	6 (2.2)	0.625
Bilirubin (in µmol/L)		15.2, 9 (4–100)	0 (0)		21.5, 9 (3–445)	11 (4.0)	0.902
Creatinine (in µmol/L)		71.1, 67 (46–102)	0 (0)		70.5, 66 (39–165)	6 (2.2)	0.598
Primary sclerosing cholangitis	2 (9.5)		0 (0)	7 (2.5)		2 (0.7)	0.128
**Surgical details**							
Operating times (in min)		183.1, 175 (82–314)	0 (0)		205.5, 190 (67–780)	4 (1.4)	0.453
Hepatic pedicle clamping	16 (76.2)		3 (14.3)	215 (77.9)		23 (8.3)	1.000
Hepatic pedicle clamping (in min)		16.5, 17.5 (0–34)	3 (14.3)		22.8, 22 (0–110)	24 (8.7)	0.066
Lymphadenectomy	13 (61.9)		0 (0)	180 (65.2)		0 (0)	0.814
Number of lymph nodes removed		4.5, 3 (1–15)	8 (38.1)		5.1, 4 (1–23)	97 (35.1)	0.535
Major hepatectomy	14 (66.7)		0 (0)	229 (83.0)		0 (0)	0.077
Extended hepatectomy	1 (4.8)		0 (0)	101 (36.6)		0 (0)	**0.003**
Perihilar bile duct resection	5 (23.8)		0 (0)	53 (19.2)		0 (0)	0.574
Vascular resection	0 (0)		0 (0)	16 (5.8)		0 (0)	0.614
Extrahepatic resection	0 (0)		0 (0)	18 (6.5)		0 (0)	0.626
Intraoperative transfusion	6 (28.6)		0 (0)	135 (48.9)		6 (2.2)	0.071
Intraoperative units of PRBC		0.9, 0 (0–5)	0 (0)		2.1, 1 (0–17)	6 (2.2)	0.051
**Histopathological details**							
Tumor size (in cm)		2.2, 2.5 (0.5–3.0)	0 (0)		7.9, 7.1 (0.9–21.0)	1 (0.4)	**<0.001**
Multifocality	0 (0)		0 (0)	106 (38.4)		0 (0)	**0.001**
Vascular invasion	1 (4.8)		7 (33.3)	66 (23.9)		102 (37.0)	**0.020**
N1	5 (23.8)		8 (38.1)	80 (29.0)		96 (34.8)	0.777
M1	0 (0)		0 (0)	10 (3.6)		0 (0)	1.000
G > 2	0 (0)		0 (0)	83 (30.1)		5 (1.8)	**0.005**
R1	1 (4.8)		1 (4.8)	45 (16.3)		7 (2.5)	0.218

Abbreviations: M.v., missing values; BMI, body mass index; ASAT, aspartate transaminase; PRBC, packed red blood cells; N, lymph node metastasis; M, distant metastasis; G, grading; R, resection margin Bold values indicate statistical significance.

**Table 2 jcm-10-04073-t002:** Number of cases with intrahepatic cholangiocarcinoma (ICC) resected over the analyzed period of 25 years.

Time Frame	Total Cases Resected (*n* = 297)	“Very early” ICC Resected (*n* = 21)	“Advanced” ICC Resected (*n* = 276)
1996–2000	57 (19.2%)	0 (0%)	57 (20.7%)
2001–2005	43 (14.5%)	7 (16.3%)	36 (13.0%)
2006–2010	72 (24.2%)	7 (9.7%)	65 (23.6%)
2011–2015	57 (19.2%)	5 (8.8%)	52 (18.8%)
2016–2021	68 (22.9%)	2 (2.9%)	66 (23.9%)

**Table 3 jcm-10-04073-t003:** Postoperative course and survival of patients with “very early” intrahepatic cholangiocarcinoma (ICC) including a comparison to a control group undergoing resection with curative intent.

Variables	“Very Early” ICC (Single Tumor ≤ 3 cm; *n* = 21)			“Advanced” ICC (*n* = 276)			*p*-Value
	*n* (%)	Mean, Median (Range)	M.v. (*n* (%))	*n* (%)	Mean, Median (Range)	M.v. (*n* (%))	
**Postoperative course**							
Postoperative transfusion	5 (23.8)		0 (0)	80 (29.0)		10 (3.6)	0.628
Postoperative units of PRBC		0.5, 0 (0–2)	0 (0)		1.6, 0 (0–29)	10 (3.6)	0.367
Postoperative complications ≥ CD3	4 (19.0)		0 (0)	97 (35.1)		2 (0.7)	0.156
Biliary leakage ISGLS Grade C	1 (4.8)		0 (0)	12 (4.3)		2 (0.7)	1.000
Hemorrhage ISGLS Grade C	1 (4.8)		0 (0)	9 (3.3)		2 (0.7)	0.528
PHLF ISGLS Grade C	0 (0)		0 (0)	15 (5.4)		2 (0.7)	0.611
ICU stay (in days)		3.2, 2 (1–13)	0 (0)		4.7, 2 (0–91)	0 (0)	0.495
Hospital stay (in days)		21.4, 20 (7–43)	0 (0)		23.2, 20 (4–95)	0 (0)	0.792
Intrahospital mortality	0 (0)		0 (0)	21 (7.6)		0 (0)	0.379
30-day mortality	0 (0)		0 (0)	13 (4.7)		4 (1.4)	0.609
90-day mortality	0 (0)		0 (0)	25 (9.1)		5 (1.8)	0.234
**Postoperative survival**							
Follow-up (in months)		66.8, 61.5 (5.9–165.8)	0 (0)		37.0, 20.8 (0.1–301.7)	0 (0)	**<0.001**
Dead at time of last follow-up	15 (71.4)		0 (0)	218 (79.0)		0 (0)	0.414
KM Survival		80.0, 62.1 (n.a.)	0 (0)		51.3, 25.3 (n.a.)	0 (0)	**0.013**
KM 1-year survival (in %)	95.2		0 (0)	70.1		0 (0)	n.a.
KM 3-year survival (in %)	79.4		0 (0)	38.5		0 (0)	n.a.
KM 5-year survival (in %)	58.2		0 (0)	24.3		0 (0)	n.a.

Abbreviations: M.v., missing values; PRBC, packed red blood cells; CD, Clavien–Dindo; ISGLS, International Study Group of Liver Surgery; PHLF, post-hepatectomy liver failure; ICU, intensive care unit; KM, Kaplan–Meier estimate; n.a., not applicable/not applied. Bold values indicate statistical significance.

## Data Availability

The dataset is available from the corresponding author on reasonable request.

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
