# Peer review of "Proposal of a New Definition of “Very Early” Intrahepatic Cholangiocarcinoma—A Retrospective Single-Center Analysis"

_jcm, 2021, doi:10.3390/jcm10184073_

Round 1
Reviewer 1 Report
Review: “Proposal of a new definition of very early intrahepatic cholangiocarcinoma- A retrospective single-center analysis”
The colleagues further describe the outcome of resections of early stage iCC in a retrospective single center cohort. Early stage was defined by tumors smaller than 3 cm. I do not have concerns regarding the data presented in this study. Description of the data as well as tables and Figure 1 are clear.
Suggestions:
-Please provide information of the recurrence rate, especially of the 21 patients with very early stage resections
Author Response
We agree that disease recurrence is a crucial endpoint, especially since the concept of a "very early" disease could also be applied with respect to transplant eligibility. Unfortunately, missing data on disease recurrence in our cohort is very high, mainly because the catchment area of patients undergoing resection at our center is large (also including international patients) and oncological aftercare and follow-up is provided decentralized by oncologists in the immediate vicinity. This hampers information flow on data regarding disease recurrence (mainly diagnosed by CT or MRI imaging). Furthermore, a timely diagnosis of disease recurrence also depends significantly on patient adherence. In summary, we believe that the overall survival provided within our work is a much more reliable/credible outcome parameter. Please also see our response to point 1.10 of the first round of revision (Reviewer #1).

Reviewer 2 Report
Overall this paper was well-written.
In my opinion, very early" ICC may include tumor location. Given the high ratio of lymph node metastasis in patients with tumor diameter <3cm, it is hardly accepted only tumor is main definition of very early disease.
Author Response
As we have pointed out critically within our Discussion section, we are well aware that a definition of a "very early" disease based solely on tumor size and number (and absence of extrahepatic disease) could lead to an underestimation of tumor aggressiveness (as reflected by the high ratio of lymph node involvement). Nonetheless, we aimed to evaluate a simplified definition by implementing variables which are easily available after a state-of-the-art preoperative tumor staging by radiological imaging (i.e. tumor size, number and extrahepatic disease). We are not sure how the location of the tumor (left lobe/right lobe; proximity to the hilar structures; proximity to central hepatic veins?) could predict tumor aggressiveness. Moreover, a classification of a specific location within the liver to statistically analyze a potential prognostic value, seems extremely difficult.

Reviewer 3 Report
Authors report a new proposal of definition of "very early" intrahepatic cholangiocarcinoma (iCC) based on reptrospective analysis of 21 patients, who undervent surgical resection. The definiton of "very early" iCC is based on tumor size and absence of extrahepatic tumor metastasis. Critical remarks: no information about histopathology or any tumor markers are given in the presented cases. No other possible prognostic factors except the size (and absence of metastasis) were studied and taken into consideration in prognosis. The agressive features of iCC depend on several other factors not only on tumor size. I suggest to include other prognostic factors and the histopathological types of the tumors into the manuscript.
Author Response
Thank you for your comment. As we have pointed out critically within our Discussion section, we are well aware that a definition of a "very early" disease based solely on tumor size (and absence of extrahepatic disease) could lead to an underestimation of tumor aggressiveness. However, preoperative tumor staging of intrahepatic cholangiocarcinoma rarely allows a more specific characterization of tumor biology, such as tumor grading or vascular invasion, or detection of lymph node metastases before surgical resection. Therefore, we aimed to evaluate a simplified definition by implementing variables which are easily available after a state-of-the-art preoperative tumor staging by radiological imaging (i.e. tumor size, number and extrahepatic disease). Nonetheless, the above-mentioned histopathological features as defined in the postoperative histopathological analyses were analyzed and are depicted in Table 1. With regard to mentioned tumor markers, a high rate of missing values of patients undergoing surgery in the early observation period significantly impairs further analyses on this specific matter.

Reviewer 4 Report
Thank you for inviting me to review this interesting paper. The topic is actual and from the paper we see how authors trying to determine prognostic value for Intrahepatic cholangiocarcinoma, especially for "very early" tumor.
This problem is still unresolved and prognostic is still undefined. Also authors, based on their experience did not answer to many questions that they asked in their paper, they defently revealed many future perspectives for studies and clinical practice.
So, this paper deserve to be published in Clinical Medicine.
But because this paper is retrospective and experience presented was collected in long term period, the limitations of the study should be mentioned in the end of the paper,
Author Response
Thank you for very much for your kind comment. We agree and have added limitations to our Discussion section, as requested (see page 8, lines 274-277).

Reviewer 5 Report
The authors offered their contribution on a definition of an early intrahepatic cholangiocarcinoma (ICC). In particular, they proposed considering (or revising) tumor size as a prognostic value, explaining the concept of a “very early” ICC.
Beetz and coll. have retrospectively taken into consideration the ICC patient cases of 25 years, studying morbidity, mortality, and long-term survival; the reviewed study has been performed by the eligibility of the patients, obtaining 297 patients.
The authors correctly point out that 1st ICC-specific classification according to the AJCC/UICC staging system (7th edition) did not include tumor size; moreover, they noted that the current (8th) edition of the AJCC/UICC staging system considers a tumor size within T1 stage.
Of course, this work contributes to the discussion inherent in the definition of a “very early” ICC, and to the development of further studies combining the traditional surgical risk factors with tumor biomarkers or mutation analyses.
However, before assuming this manuscript for publication, there are minor questions to be considered.
- The data reported in the tables, in particular 1 and 3, are not all legible and clear (i.e. n, mean, median, etc with respect to the numerical data), and the tables could be improved. Please, try to improve the alignment of the data to facilitate a more immediate reading.
- Within the discussion, the authors could apply a tone or a style more determined to strengthen their thesis. Of course, there is still a lot to discuss on a definition of early disease, but there is no doubt that the smaller the tumor size to be removed, the better the outcome, and the management of the patient.
In this regard, some suggestions could derive from different methods used to treat smaller ICC tumors, such as thermal ablation for a single ICC smaller than 3 cm, less invasive, with an overall better outcome, in agreement with very early disease (doi.org/10.1007/s00330-021-08216-x). - The authors have considered several perioperative variables, among which, interestingly, preoperative leukocyte concentrations show significantly lower levels in patients undergoing resection of “very early” ICC. Since the authors hope that the concept of “very early” ICC is considered in combination with other variables or parameters, other factors could be included. The authors could take into consideration other papers, such as doi.org/10.2147/CMAR.S317954, in which the prognostic value in ICC derived from inflammation-based biomarkers. In particular, peripheral counts of neutrophils, monocytes, and lymphocytes were integrated into a systemic inflammation response index (SIRI). Perhaps, the association of a traditional factor as the tumor size with inflammation-based biomarkers could represent a reinforcing combination.
- Regarding the tumor biomarkers as predicting prognosis, it is not a simple issue. The data are scarce, and not always so in agreement. In addition, the tumor biomarkers could show variable behavior depending on time and on tumor growth: in this regard, the tumor size as a prognostic value could assume much more consistency.
Author Response
The authors offered their contribution on a definition of an early intrahepatic cholangiocarcinoma (ICC). In particular, they proposed considering (or revising) tumor size as a prognostic value, explaining the concept of a "very early" ICC.
Beetz and coll. have retrospectively taken into consideration the ICC patient cases of 25 years, studying morbidity, mortality, and long-term survival; the reviewed study has been performed by the eligibility of the patients, obtaining 297 patients.
The authors correctly point out that 1st ICC-specific classification according to the AJCC/UICC staging system (7th edition) did not include tumor size; moreover, they noted that the current (8th) edition of the AJCC/UICC staging system considers a tumor size within T1 stage.
Of course, this work contributes to the discussion inherent in the definition of a "very early" ICC, and to the development of further studies combining the traditional surgical risk factors with tumor biomarkers or mutation analyses.
5.1 The data reported in the tables, in particular 1 and 3, are not all legible and clear (i.e. n, mean, median, etc with respect to the numerical data), and the tables could be improved. Please, try to improve the alignment of the data to facilitate a more immediate reading.
Response: First of all, thank you very much for your thorough analyses and comments on our work, which have led us to edit our Discussion section in several points. With regard to Point 5.1, we absolutely agree and have edited the mentioned tables by separating the columns for binary and metric/numerical variables (see revised Tables 1 and 3 on page 5 and 6). We hope to meet your approval.
5.2 Within the discussion, the authors could apply a tone or a style more determined to strengthen their thesis. Of course, there is still a lot to discuss on a definition of early disease, but there is no doubt that the smaller the tumor size to be removed, the better the outcome, and the management of the patient.
In this regard, some suggestions could derive from different methods used to treat smaller ICC tumors, such as thermal ablation for a single ICC smaller than 3 cm, less invasive, with an overall better outcome, in agreement with very early disease (doi.org/10.1007/s00330-021-08216-x).
Response: Although we agree with the general assumption that a smaller tumor is associated with better outcome, aim of recent publications (and our own) was to define or evaluate an appropriate cut-off (and hence a definition of "very early" disease) for therapeutic decision making (such as liver transplantation), especially since Hyder et al. were able to demonstrate that tumor size loses further prognostic value at a certain point (approximately 7 cm) and others failed to demonstrate tumor size as independent prognostic factor, as pointed out in our Discussion section.
However, we have now commented on the prognostic value of a 3 cm tumor size cut-off also for interventional approaches such as thermal ablation and added the above-mentioned citation by Kim et al. to our Discussion section (see page 7, lines 227-229).
5.3 The authors have considered several perioperative variables, among which, interestingly, preoperative leukocyte concentrations show significantly lower levels in patients undergoing resection of "very early" ICC. Since the authors hope that the concept of "very early" ICC is considered in combination with other variables or parameters, other factors could be included. The authors could take into consideration other papers, such as doi.org/10.2147/CMAR.S317954, in which the prognostic value in ICC derived from inflammation-based biomarkers. In particular, peripheral counts of neutrophils, monocytes, and lymphocytes were integrated into a systemic inflammation response index (SIRI). Perhaps, the association of a traditional factor as the tumor size with inflammation-based biomarkers could represent a reinforcing combination.
Response: This is indeed an interesting aspect and idea calling for regression analyses including evaluation of potential multiplicative or additional factor interactions in order to design a prognostic score based on above-mentioned variables. Given the focus of the manuscript with a more descriptive approach of a "very early" disease and the short amount of time available for a thorough revision of the manuscript, we believe that this issue should be addressed in the near future. Accordingly, we have commented on this issue within our Discussion section and added the recent publication by Jin et al., as suggested (see page 8, lines 246-247).
5.4 Regarding the tumor biomarkers as predicting prognosis, it is not a simple issue. The data are scarce, and not always so in agreement. In addition, the tumor biomarkers could show variable behavior depending on time and on tumor growth: in this regard, the tumor size as a prognostic value could assume much more consistency.
Response: We agree and have therefore edited our deliberations on this point in our Discussion section (see page 8, lines 259-262).

This manuscript is a resubmission of an earlier submission. The following is a list of the peer review reports and author responses from that submission.
Round 1
Reviewer 1 Report
Whyt in advanvace intrahepatic cholangiocarcinoma the bilirubine level is inferior the very early, even if is not statistically significative, maybe the authors can explain better this results
Reviewer 2 Report
This single-center, retrospective study was evaluating the outcome of patients who underwent liver resection for ICC depending on tumor size. The authors studied cohort of 298 patients over the 25 years period. They found that the outcome of “very early” ICC was not significantly better that in “early” ICC.
I read the manuscript with interest and I have several comments to the authors:
The main problem with this study is the time span of 25 years, where only 298 patients were treated. This might create selection bias. Have you tried to divide you cohort into two groups according to the resection era? Have you seen any trend that very early ICC were operated recently, due to better quality of diagnostic tools? 1
- “Very early” ICC group is small (n=12), so statistical comparison might with two others group is difficult.
- What was the reason for major hepatectomies in “very early” ICC (2/3 of patients)? Why not smaller resections?
- What is the explanation for high percentage of intraoperative blood transfusion (41.7%)?
- What is your policy regarding lymphadenectomy during resection of ICC, as there were different lymphadenectomy rates in your three groups?
- How many patients with ICC had underlying liver disease (i.e PSC)?
- What was the causes of in-hospital mortality in your cohort?
- The last paragraph in section 3.4 is confusing. Was there a trend toward or significantly better survivor better survival in “very early” ICC compared to “early” and “advanced” ICC? Please clarify this.
- Do you have any information about recurrence in your cohort?